# Multi-Scenario Prediction Analysis of Carbon Peak Based on STIRPAT Model-Take South-to-North Water Diversion Central Route Provinces and Cities as an Example

Qingxiang Meng, Baolu Li, Yanna Zheng, Huimin Zhu, Ziyi Xiong, Yingchao Li * and Qingsong Li *

College of Resources and Environment, Henan Agricultural University, Zhengzhou 450046, China; qxmeng@henau.edu.cn (Q.M.); baoluli2023@stu.henau.edu.cn (B.L.); yannazheng2023@stu.henau.edu.cn (Y.Z.); zhm@stu.henau.edu.cn (H.Z.); xiongziyi@stu.henau.edu.cn (Z.X.)
* Correspondence: ycli666@henau.edu.cn (Y.L.); qs.li@henau.edu.cn (Q.L.)

**Abstract:** With the increase in energy demand, environmental issues such as carbon emissions are becoming more and more prominent. China will scale its intended nationally determined contributions by adopting more vigorous policies and measures. China aims to have $CO_2$ emissions peak before 2030 and achieve carbon neutrality before 2060. The current challenge and priority of China's high-quality development is to ensure a harmonious balance between the ecological environment and the economy. The South-to-North Water Diversion Project passes through Beijing, Tianjin, Henan, and Hebei, which were chosen as the study sites. The carbon emission data was from the China Carbon Emission Database 2000–2019. Decoupling modeling using statistical yearbook data from four provinces and municipalities. KMO and Bartlett's test used SPSS 27 software. The selection of indicators was based on relevance. Analyses were performed using the extended STIRPAT model and ridge regression. Moreover, projections of carbon peaks in the study area for 2020–2035 under different rates of change were simulated by the scenario analysis method. The results show that: (1) The decoupling analysis of the four provinces and cities from 2000-2019 gradually shifts to strong decoupling; (2) Resident population, energy structure, and secondary industry as a proportion of GDP significantly impact carbon emissions; (3) From 2000–2035, Beijing and Henan experienced carbon peaks. The peak time in Beijing was 96.836 million tons in 2010. The peak time in Henan was 654.1004 million tons in 2011; (4) There was no peak in Hebei from 2000–2035.

**Keywords:** carbon peaking; decoupling; South-to-North Water Diversion Project; STIRPAT mode

## 1. Introduction

Against the backdrop of a slowdown in total world carbon emissions, there are significant differences between the developed economy's and the emerging economy's current carbon emissions status and the emissions outlook. Research on carbon emissions has become the attention of scholars at home and abroad [1,2]. The successive establishment of the United Nations Framework Convention on Climate Change, the Kyoto Agreement, and the Paris Agreement demonstrate the international community's collaborative endeavors toward worldwide low-carbon progress. Their ultimate goal is to reduce emissions and save energy, controlling the temperature's rise. In order to achieve green development, it is necessary to study issues such as carbon emissions and their influencing factors [3]. China has made ecological progress an essential part of its 13th Five-Year Plan, implementing the development concepts of innovation, coordination, greenness, openness, and sharing through scientific, technological, and institutional innovation. This includes the implementation of an optimized industrial structure and the building of a low-carbon energy system. Developing green buildings and low-carbon transport establishment of a national carbon emissions trading market and a series of other policy measures will form a new pattern of modernization and construction for the harmonious development of humanity

and nature [4,5]. China is aiming to reach the highest point of its carbon dioxide emissions by 2030 and to become carbon neutral by 2060.

Scholars worldwide have utilized the STIRPAT model to explore the trajectories and peak periods of HCEs in 30 provinces in China until 2040 and have formulated three distinct scenarios (baseline, low, and high) to predict carbon peaks. The findings indicate that in at least one of the scenarios, 25 provinces have the potential to achieve peak HCE by 2030, whereas five provinces would fall short of meeting the 2030 emissions target [6]. Changes in carbon emissions also require more certainty due to uncertainty about future development patterns, making meeting peak targets challenging. Taking Shandong, Henan, and Guangdong as three of China's most populous provinces as examples, the effects of uncertainty in carbon accounting principles, drivers, and simulation mechanisms on achieving peak targets were analyzed [7]. They used the LMDI method to decompose and analyze the driving factors affecting China's carbon dioxide emissions by studying the detailed situation of 41 sub-industries from 2000 to 2016. Based on various official policies and documents, the carbon intensity reduction potential for 2020 and 2030 was predicted [8]. This paper investigates carbon emission peaks in China based on a comparative analysis of energy transition in China and the United States [9]. A novel multifactor decomposition method for carbon emissions is proposed [10]. Multifactor decomposition models based on the Kaya Identity extension and the LMDI decomposition methodology from energy, economic, and social perspectives provide quantitative results. On this basis, an evaluation system was constructed by applying the entropy weight method, and the carbon emission indices of six power generation modes in China were generated from three dimensions: environment, energy, and economy. It also established a carbon emission dynamic model based on the carbon emission data of the past 40 years and, combined with Tapio's decoupling theory, predicted China's carbon emissions under multiple scenarios for the next 40 years. Based on the carbon emission panel data of countries along the "Belt and Road" from 1970 to 2018 and the environmental Kuznets curve (EKC) theory, a panel model was established for each country group for research [11]. The study proposes that carbon substitution, carbon emission reduction, carbon sequestration, and carbon recycling are the four main ways to achieve carbon neutrality, of which carbon substitution will be the backbone of carbon neutrality [12]. According to the high, medium, and low scenarios, China's carbon emissions are projected to fall to $22 \times 10^8$, $33 \times 10^8$ and $44 \times 10^8$ tons in 2060, respectively. Seven implementation recommendations are made for China to achieve carbon neutrality. The results show that the earlier the time of peak carbon emissions, the more significant the economic impact on China; under the three scenarios of peak carbon emissions, government revenues and savings have a significant decline, and the rest of the economic indicators do not cause too much impact; the impact of peak carbon emissions on the output of the construction industry is small, and the output of the other sectors has a slight increase [13]. Focusing on natural resource management under the "dual-carbon" objective, nine experts proposed innovative natural resource management strategies from different perspectives, providing reference and reference for the construction of a low-carbon oriented natural resource management system based on the multi-level perspective of "resource elements-territorial space-ecosystem". This provides a reference for constructing a low-carbon-oriented natural resource management system based on the multi-level perspective of "resource elements, land space, and ecosystem". [14].

In summary, there is still some controversy among many scholars as to whether China can achieve the goals of carbon peaking and carbon neutrality on time, and there are some areas for improvement in the analysis of carbon peaking at the basin level [15]. This study uses the provinces and cities through which the South-to-North Water Diversion Project passes as the study area to analyze, build models, and carry out prediction analyses to provide the government with corresponding carbon peak strategies.

## 2. Research Area

The four provinces and cities of Beijing, Tianjin, Hebei, and Henan, through which the central line of the South-to-North Water Diversion passes, were used for the study (Figure 1). The provinces and cities through which the South-to-North Water Diversion passes have a significant impact on China's socio-economic development. In the meantime, "Beijing–Tianjin–Hebei" has emerged as the third most significant economic district, following "The Yangtze River Delta" and "Pearl River Delta". The effectiveness of its carbon emission reduction is directly related to the achievement of China's carbon emission reduction targets [16–19]. The South-to-North Water Diversion Mainline Project, an essential part of the National South-to-North Water Diversion Project, is a major strategic infrastructure built to alleviate the severe shortage of water resources in China's Huanghua Hai Plain and optimize the allocation of water resources and is a century-long project related to the sustainable economic and social development of the receiving areas in the provinces and cities of Henan, Hebei, Tianjin, and Beijing and the well-being of the future generations. The regional scope of the South-to-North Water Diversion Project is geographically vast. The spatial differences in each region's resources, population, economy, and industrial structure are apparent, and the specific measures for regional carbon emission reduction are different. Research on land use carbon emissions and influencing factors in the region provide a theoretical and practical basis for its emission reduction.

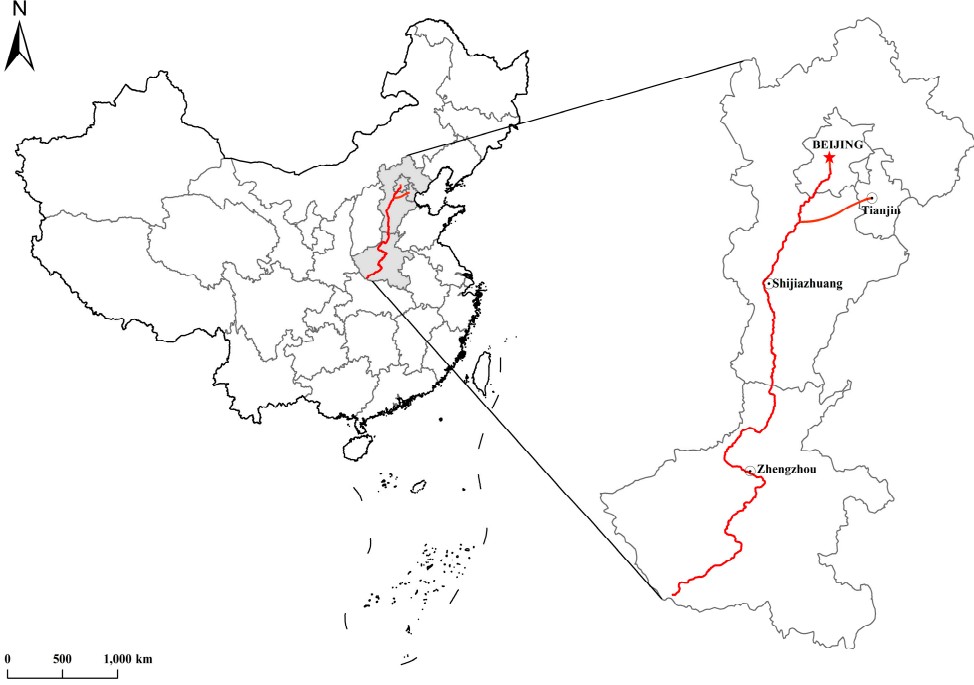

**Figure 1.** Provinces and cities passing through the South-to-North Water Diversion Central Route.

## 3. Research Methods and Data

### 3.1. Analysis of Carbon Emissions

From 2000–2019, Beijing's carbon dioxide emissions have been characterized by a "U" shape, beginning with an increase and then a decrease. The amount increased from 63,471,900 tons in 2000 to its highest point of 96,836,000 tons in 2010 and has since decreased to 70,611,800 tons in 2019. From 2000–2019, Tianjin has seen a steady increase in its $CO_2$ emissions, with a slight peak from 2010 to 2013 and a total of 151,032,500 tons in 2013. Hebei's carbon dioxide emissions rose steadily from 2000–2019, with the most significant surge occurring between 2000 and 2013. The total amount of $CO_2$ released in 2013 amounted to 657.72 million tons. From 2014–2017, there was a slight decrease, which kept rising. From 2000–2019, Henan experienced a fluctuating amount of $CO_2$ emissions.

There was an upward trend from 2000–2011 and a downward trend from 2012–2019. The total amount of $CO_2$ released in 2019 amounted to 463,998,300 tons (Figure 2).

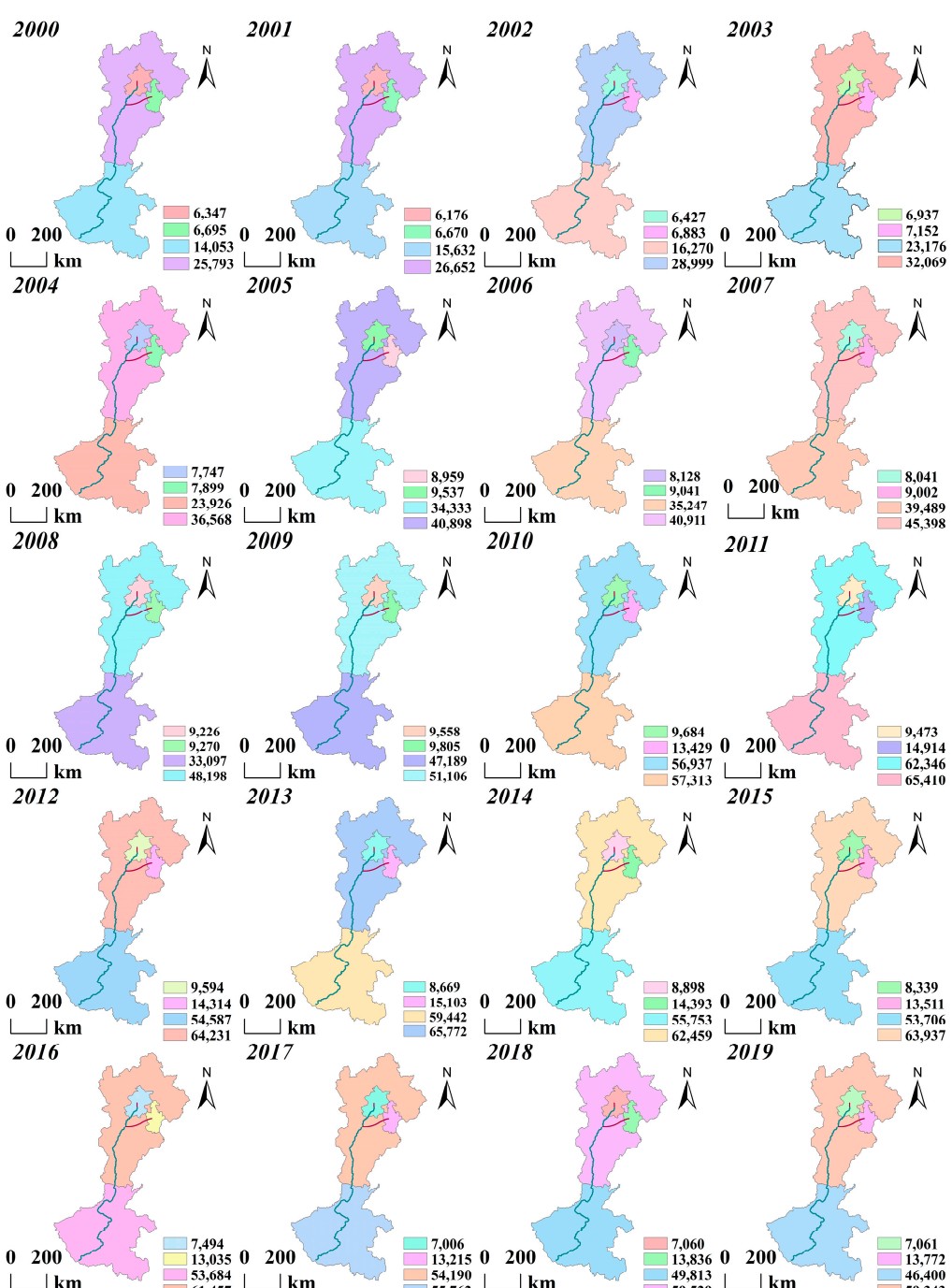

**Figure 2.** Carbon emissions from Beijing, Tianjin, Hebei, and Henan, 2000–2019 ($10^4$ t).

Among the four provinces and cities of Beijing, Tianjin, Hebei, and Henan in the period of 2000–2019, Beijing and Tianjin's annual carbon dioxide emissions were relatively small. Hebei's carbon dioxide emissions commonly remained high, and only in 2011, 2012, and 2018 did Henan's carbon dioxide emissions exceed those of Hebei, which shows that Hebei is a large province in terms of carbon dioxide emissions.

*3.2. Research Methodology*

3.2.1. Decoupling Model

The concept of decoupling was introduced by the Organization for Economic Cooperation and Development in 2002. This concept describes models and demonstrates the correlation between economic growth and environmental quality impairment [20]. Tapio proposed an improved decoupling model in 2005. The decoupling analysis method has been widely used in major industries, and this study uses decoupling analysis to study the relationship between economic growth and carbon emissions, as shown in Equation (1).

$$MI_{t2-t1} = \frac{\%\Delta CE}{\%\Delta GDP} = \frac{\dfrac{CE_{t2} - CE_{t1}}{CE_{t1}}}{\dfrac{GDP_{t2} - GDP_{t1}}{GDP_{t1}}} \tag{1}$$

In Equation (1), $MI_{t2-t1}$ is the decoupling index from moment $t1$ to moment $t2$, $CE_{t2}$ and $CE_{t1}$ are the carbon emissions at $t1$ and $t2$. $GDP_{t2}$ and $GDP_{t1}$ are GDP at moments $t1$ and $t2$.

They use elasticity values of 0, 0.8, and 1.2 as thresholds to account for the decoupling of $CO_2$ from economic growth. For example, when the elasticity value is less than 0, $CO_2$ shows strong negative or strong decoupling. Therefore, the status and degree of $CO_2$ decoupling must be determined according to the situation. The specific indicator system [21–23] is shown in Table 1.

**Table 1.** Tapio Decoupling Indicator System.

| Status | Decoupling State | ΔCE | ΔGDP | MI |
|---|---|---|---|---|
| Decoupling | Strong decoupling | <0 | >0 | MI < 0 |
| | Weakly decoupling | >0 | >0 | 0 < MI ≤ 0.8 |
| Negative decoupling | Recessive decoupling | <0 | <0 | MI > 1.2 |
| | Strong negative decoupling | >0 | <0 | MI < 0 |
| | Weak negative decoupling | <0 | <0 | 0 < MI ≤ 0.8 |
| | Expansion negative decoupling | >0 | >0 | MI > 1.2 |
| Connection | Expansion connection | >0 | >0 | 0.8 < MI ≤ 1.2 |
| | Recession connection | <0 | <0 | 0.8 < MI ≤ 1.2 |

3.2.2. STIRPAT Model

The IPAT equation proposed by Western scholars in the 1970s is a classic model for exploring the relationship between economic growth and energy consumption [24,25]. Its expression is shown in Equation (2).

$$I = P \times A \times T \tag{2}$$

In Equation (2), *I*, *P*, *A*, and *T* represent environmental conditions, population, economic affluence, and technology level, respectively. However, the IPAT equation only considers the relationship between economic development and environmental pressure on energy consumption and sets it as a simple linear relationship with certain limitations. Therefore, in this paper, the STIRPAT model obtained by extending the IPAT model was chosen, and the model's expression is shown in Equation (3).

$$I = aP^b A^c T^d e \tag{3}$$

In Equation (3): *a* is the model coefficient; *b, c, d* are prognostic factors; *e* is the error factor.

### 3.3. Data Sources

Carbon emissions from the four provinces and cities of Beijing, Tianjin, Hebei, and Henan in 2000–2019 were sourced from the China Carbon Emissions Database (CEADs) [26–29]. Resident population, GDP per capita, energy consumption intensity (energy consumption/GDP), energy structure (coal consumption/total energy consumption), and secondary industry share of GDP are from the China Energy Statistical Yearbook and China Statistical Yearbook (2000–2019).

## 4. Analysis of Results

### 4.1. Decoupling Model Analysis

Carbon decoupling refers to the issue of the relationship between changes in $CO_2$ emissions and economic growth. When economic growth is achieved at the same time that $CO_2$ emissions grow at a negative rate or a rate less than the economic growth rate, it can be regarded as decoupling, which is essentially a measure of whether economic growth comes at the cost of resource consumption and environmental damage. Through the four provinces and cities through which the South-to-North Water Diversion Project passes, from 2000–2019, carbon emission and GDP data were calculated according to the decoupling model, and the decoupling was calculated as shown in (Figure 3).

| | 2000-2001 | 2001-2002 | 2002-2003 | 2003-2004 | 2004-2005 | 2005-2006 | 2006-2007 | 2007-2008 | 2008-2009 | 2009-2010 |
|---|---|---|---|---|---|---|---|---|---|---|
| **Beijing** | V | II | II | II | VI | I | I | VII | II | II |
| **Tianjin** | I | II | II | II | II | II | I | II | II | II |
| **Hebei** | II | IV | II | II | II | II | II | II | VI | II |
| **Henan** | VII | II | VI | II | VI | II | II | I | VI | VII |

| | 2010-2011 | 2011-2012 | 2012-2013 | 2013-2014 | 2014-2015 | 2015-2016 | 2016-2017 | 2017-2018 | 2018-2019 | |
|---|---|---|---|---|---|---|---|---|---|---|
| **Beijing** | I | II | I | VI | I | I | I | II | II | |
| **Tianjin** | IV | I | II | I | I | I | II | VI | V | |
| **Hebei** | VI | II | II | I | VI | I | I | VI | V | |
| **Henan** | VII | I | VII | I | I | I | II | I | I | |

| **Decoupling status** | Strong decoupling | | | I | Weak negative decoupling | | | | | V |
| | Weakly decoupling | | | II | Expansion negative decoupling | | | | | VI |
| | Strong negative decoupling | | | IV | Expansion Connection | | | | | VII |

**Figure 3.** Dynamic relationship between carbon emissions and economic growth in four provinces and cities from 2000–2019.

Beijing from 2000–2009 experienced a weak negative decoupling, weak decoupling, expansion negative decoupling, and a strong decoupling stage for a better trend, economic growth, and carbon emissions among the more reasonable. Decoupling mainly alternated between weak and strong decoupling from 2010–2019. Under continuous economic acceleration, the economic growth rate was greater than the growth rate of carbon emissions, indicating that energy conservation and emission reduction efforts achieved the intended results.

Tianjin's carbon emissions and economic development in the 2000–2009 timeframe were mainly weakly decoupling, with a large amount of carbon emitted mainly from the secondary industry, and the growth rate of carbon emissions was greater than that of economic growth, leading to excessive carbon emissions. Expansion of negative decoupling occurred from 2009–2010, and strong decoupling predominated from 2010–2019, indicating significant results in energy conservation, emission reduction, and industrial restructuring. The relationship between carbon emissions and economic growth was within a more reasonable range.

Hebei was mainly weakly decoupled from 2000–2009, closely related to Hebei's energy emissions. Hebei is a traditional industrial province and China's number one iron and steel province, resulting in a continuous increase in carbon emissions. Strong decoupling occurred twice from 2010–2019, in 2016 and 2017, respectively. This demonstrates that Hebei promotes the transformation of old and new kinetic energy, accelerates the pace of transformation and upgrading of traditional industries, and accelerates the development of new industries. It also shows that the industry's momentum toward the middle and

high end is vital. It is necessary to deeply promote the manufacturing industry's high-end, intelligent, and green development.

Henan was mainly dominated by weak decoupling and expansion negative decoupling from 2000–2009, with weak decoupling occurring five times and expansion negative decoupling occurring three times, indicating an irrational match between carbon emissions and economic growth. Strong decoupling occurred six times from 2010–2019. Then, there was a better match between economic growth and carbon emissions, with economic growth being more significant than the rate of carbon emissions and a sustained reduction in carbon emissions. This is closely related to the government's energy-saving and emission-reduction policies, industrial restructuring, and green development. The relationship between economic growth and carbon emissions has been gradually harmonized to promote high-quality development in Henan.

*4.2. STIRPAT Model Regression Fit Analysis*

4.2.1. Description of Variables

The following is a collection and collation of basic data from four provinces and municipalities: Beijing, Tianjin, Hebei, and Henan carbon emissions, 2000–2019, permanent population, GDP per capita, energy consumption intensity (energy consumption/GDP), and energy structure (coal consumption/total energy consumption). This includes data on the share of the secondary sector in GDP. The maximum, minimum, mean, and standard deviation of each variable after taking the logarithm are shown in (Table 2).

**Table 2.** Description of variables.

| Variant | Abbreviated Symbol | Minimum Value | Maximum Value | Average Value | Standard Deviation |
|---|---|---|---|---|---|
| Carbon emissions | I | 617.61 | 6577.2 | 2754.21 | 2115.56 |
| Permanent population | P | 1001.14 | 9901 | 4937.08 | 3550.22 |
| GDP per capita | A | 5450 | 161,776 | 45,876.44 | 35,365.7 |
| Intensity of energy consumption | T | 0.21 | 2.43 | 1.06 | 0.58 |
| Energy mix | Y | 0.02 | 1.3 | 0.81 | 0.33 |
| Share of secondary sector in GDP | Z | 16.2 | 60.1 | 44.4 | 11.93 |

4.2.2. Variable Correlation Analysis

The base data of the four provinces and cities were logarithmically processed, and factor analyses of the variables by SPSS 27 software, KMO, and Bartlett's test are shown in Table 3. The range of values between 0.7 and 0.8 is barely suitable, and between 0.8 and 0.9 is suitable. The values are 0.8 for Beijing, 0.8 for Tianjin, 0.6 for Hebei, and 0.7 for Henan, all with a significance of 0.000.

**Table 3.** KMO and Bartlett's test.

| | | Beijing | Tianjin | Hebei | Henan |
|---|---|---|---|---|---|
| KMO Sampling Suitability Measure | | 0.8 | 0.6 | 0.6 | 0.7 |
| Bartlett's Sphericity test | Approximate Chi-squared value | 183.40 | 238.29 | 210.19 | 186.54 |
| | (number of) degrees of freedom (physics) | 15 | 15 | 15 | 15 |
| | Significance | 0.00 | 0.00 | 0.00 | 0.00 |

After the KMO and Bartlett tests, the indicators that best reflect the input–output relationship were screened using Pearson correlation coefficients (Figure 4). The closer the absolute value of the Pearson correlation coefficient is to 1, the stronger the correlation between the indicator and carbon emissions.

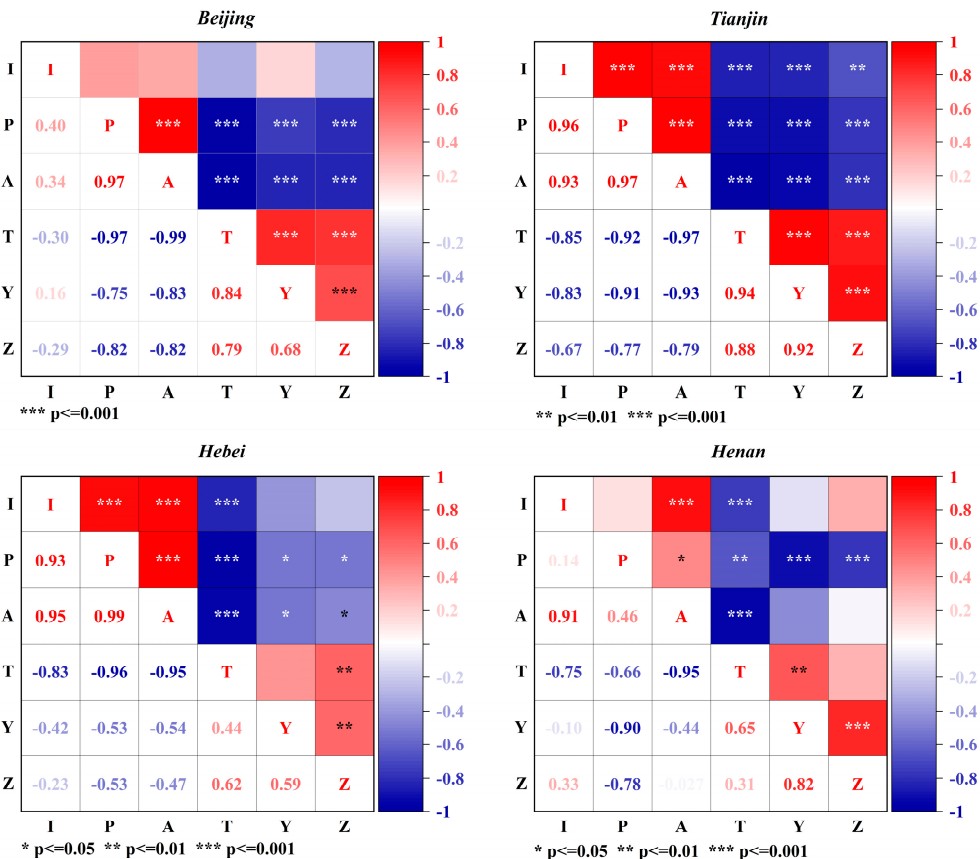

**Figure 4.** Pearson correlation between indicators.

There is a strong positive correlation of 0.98 between the resident population and GDP per capita in Beijing and a strong negative correlation with energy consumption intensity, energy structure, and the proportion of secondary industry in GDP, which are −0.97, −0.75, and −0.82, respectively. There is a strong negative correlation between per capita GDP and energy consumption intensity, energy structure, and the proportion of the secondary industry. Carbon emissions in Tianjin positively correlate with the resident population and per capita GDP, up to 0.96. A negative correlation exists between permanent population and energy consumption intensity, energy structure, and the proportion of secondary industry in GDP. In Hebei Province, carbon emissions have a strong positive correlation with permanent population and per capita GDP, reaching the highest, 0.95, and a negative correlation with energy consumption intensity, energy structure, and the proportion of secondary industry in GDP. The positive correlation between carbon emission and per capita GDP in Henan Province is 0.91, and the negative correlation between carbon emission and energy consumption intensity and energy structure is 0.91.

### 4.3. Tapio Decoupling Model Analysis

According to the basic form of the STIRPAT model, the extended STIRPAT model was constructed, and its expression is shown in Equation (4).

$$I = aP^b A^c T^d Y^i Z^j e \tag{4}$$

In order to facilitate subsequent data analysis and processing, Equation (4) is logarithmic to obtain Equation (5):

$$lnI = blnP + clnA + dlnT + ilnY + jlnZ + lne \tag{5}$$

In formula (5), *b*, *c*, *d*, *i*, and *j* are prediction coefficients representing the amount of change in carbon emissions *b*%, *c*%, *d*%, *i*%, and *j*% that can be induced by a 1% change in the resident population (10 thousand people), GDP per capita (RMB/person), energy consumption intensity (10 thousand tons of standard coal/billions), energy structure (tons of standard coal/tons), and the share of secondary industry in GDP (%). The regression model is used to construct a carbon emission prediction model for the four provinces and municipalities of the South-to-North Water Diversion Project and then to analyze carbon peaking by predicting the future carbon emission trends of the four provinces and municipalities based on the baseline scenario, the green scenario, and the high-speed scenario.

### 4.4. Ridge Regression Results

The results of the SPSS 27 software test showed that the variance inflation factor between the variables was higher than 10. In order to avoid multicollinearity between the influencing factors, ridge regression analysis was used to fit the carbon emissions to the influencing factors, and the carbon emission prediction models were constructed separately for the study area. Individual carbon emission forecasting models were created for the research region. The pertinent results can be found in Table 4.

**Table 4.** Carbon emission regression fitting results for four provinces and cities of South-to-North Water Diversion Central Route Project.

| Province | P | A | T | Y | Z | Constant | k | $R^2$ |
|---|---|---|---|---|---|---|---|---|
| Beijing | 0.2561 | 0.1878 *** | −0.0467 | 0.1826 *** | −0.0601 | 5.3592 *** | 0.05 | 0.73 |
| Tianjin | 1.0539 *** | 0.1522 *** | −0.0229 | −0.0764 | 0.1613 | −0.5087 | 0.15 | 0.91 |
| Hebei | 3.3382 *** | 0.2347 *** | −0.1063 ** | −0.2363 | 1.4690 *** | −26.7293 *** | 0.10 | 0.96 |
| Henan | −0.1369 | 0.3432 *** | −0.3617 *** | 0.2525 | 2.3362 *** | −0.7557 | 0.15 | 0.93 |

Note: ***, ** represent $p < 0.01$ and $p < 0.05$, respectively.

The resulting carbon emission projection models for Beijing, Tianjin, Hebei, and Henan are shown in Equations (6)–(9).

$$\ln I = 0.2561\ln P + 0.1878\ln A - 0.0467\ln T + 0.1826\ln Y - 0.0601\ln Z + 5.3592 \tag{6}$$

$$\ln I = 1.0539\ln P + 0.1522\ln A - 0.0229\ln T - 0.0764\ln Y + 0.1613\ln Z - 0.5087 \tag{7}$$

$$\ln I = 3.3382\ln P + 0.2347\ln A - 0.1063\ln T - 0.2363\ln Y + 1.4690\ln Z - 26.7293 \tag{8}$$

$$\ln I = -0.1369\ln P + 0.3432\ln A - 0.3617\ln T + 0.2525\ln Y + 2.3362\ln Z - 0.7557 \tag{9}$$

According to Equations (6)–(9), the substitution of the data to obtain the carbon emissions and the projected carbon emissions of the four regions of Beijing, Tianjin, Hebei, and Henan from 2000–2019 is shown in (Figure 5).

### 4.5. Peak Carbon Scenario Projections

Based on the extended STIRPAT model, a forecasting study was conducted with 2020 as the base year and 2035 as the end year. In 2020, due to the impact of "new coronavirus pneumonia", the scenarios were based on the rates of change in the 13th Five-Year Plan, as shown in Tables 5–8. The change characteristics for 2021–2035 were divided into three time periods: 2021–2025, 2026–2030, and 2031–2035, with each period set at five years.

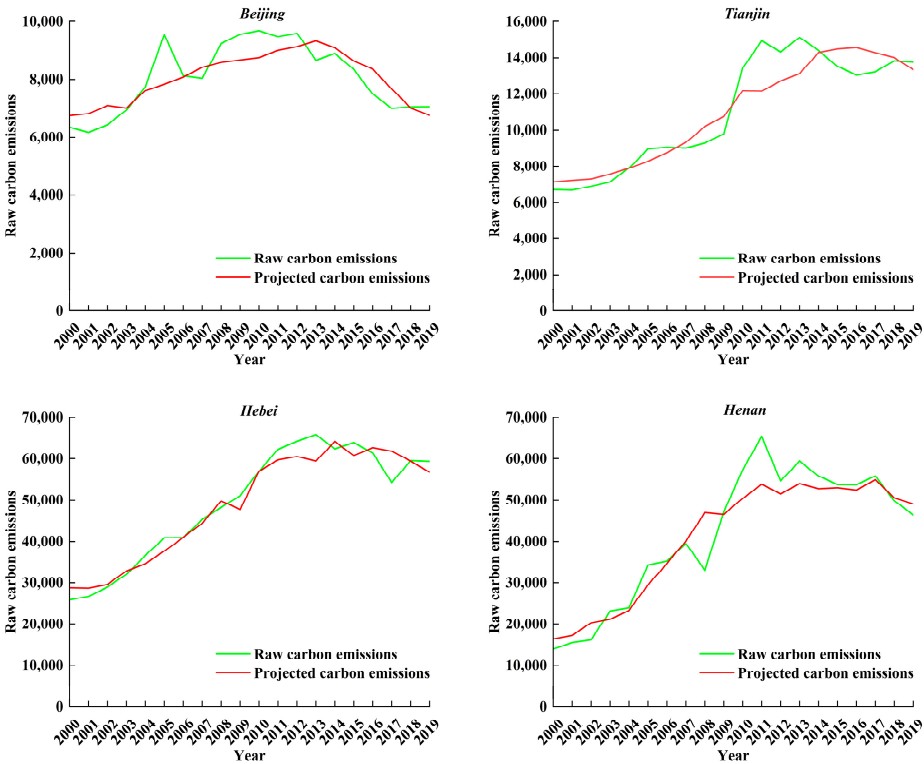

**Figure 5.** Actual carbon emissions and projections for four provinces and cities, 2000–2019 ($10^4$t).

### 4.5.1. Basis for Setting Indicators in Beijing

The setting of various indicators in Beijing is shown in Table 5. The population setting was based on the Beijing Urban Master Plan (2016–2035), which states that the resident population of Beijing will be within 23 million in 2035, and the resident population of Beijing will be 21.9 million in 2019, which allowed us to calculate that the average annual rate of change from 2019–2035 will be controlled to be around 0.3139%. For 2021, the low speed of the population can be set at 0.2800%, the medium speed at 0.3100%, and the high speed at 0.3400%, with a constant annual decline of 0.001% during the forecast period. Per capita GDP was based on the 14th Five-Year Plan for Beijing's National Economic and Social Development and the outline (draft) of the long-term goals for 2035. By 2035, per capita GDP will reach more than 320,000 yuan, and the city's comprehensive competitiveness will rank among the highest in the world. In 2019, the per capita GDP of Beijing was 161,776 yuan. It can be calculated that the average annual change rate from 2019–2035 will be controlled at around 6.1128%. It can be set that the per capita regional gross domestic product in 2021 will decrease at a low speed of 5.9100%, a medium speed of 6.1128%, and a high speed of 6.3100%. During the forecast period, the annual average decline will be 0.28%. The energy consumption intensity in Beijing was 0.46% in 2010 and 0.21% in 2019. Therefore, the change rate between 2010 and 2019 can be calculated to be −5.52%. The energy consumption intensity in 2021 can be set as −5.8200% at low speed, −5.5200% at medium speed, and −5.2200% at high speed. The energy structure in Beijing was 0.38% in 2010 and 0.02% in 2019. Therefore, the change rate between 2010 and 2019 can be calculated to be −9.34%. It can be set that the energy structure in 2021 will have a low speed of −9.5400%, a medium speed of −9.3400%, and a high speed of −9.1400%. In Beijing, the proportion of the secondary industry in GDP was 24% in 2010 and 16.2% in 2019, so the change rate between 2010 and 2019 can be calculated as −3.25% and the proportion of the secondary industry in GDP in 2021 can be set as −3.5500% at low speed, −3.2500% at medium speed and −3.0550% at high speed.

**Table 5.** Beijing's indicator settings.

| Rate of Change | Vintages | Rate of Change Setting | | | | |
| --- | --- | --- | --- | --- | --- | --- |
| | | Permanent Population | GDP per Capita | Intensity of Energy Consumption | Energy Mix | Share of Secondary Sector in GDP |
| Low | 2019–2020 | −0.0569% | 7.7771% | −5.7000% | −19.9300% | −3.9744% |
| | 2021–2025 | 0.2800% | 5.9100% | −5.8200% | −9.5400% | −3.5500% |
| | 2026–2030 | 0.2750% | 4.5100% | −5.8250% | −10.0400% | −3.5550% |
| | 2031–2035 | 0.2700% | 3.1100% | −5.8300% | −10.5400% | −3.5600% |
| Middle | 2019–2020 | −0.0569% | 7.7771% | −5.7000% | −19.9300% | −3.9744% |
| | 2021–2025 | 0.3100% | 6.1128% | −5.5200% | −9.3400% | −3.2500% |
| | 2026–2030 | 0.3050% | 4.7128% | −5.5250% | −8.8400% | −3.2550% |
| | 2031–2035 | 0.3000% | 3.3128% | −5.5300% | −8.3400% | −3.2600% |
| High | 2019–2020 | −0.0569% | 7.7771% | −5.7000% | −19.9300% | −3.9744% |
| | 2021–2025 | 0.3400% | 6.3100% | −5.2200% | −9.1400% | −3.0500% |
| | 2026–2030 | 0.3350% | 4.9100% | −5.2250% | −8.6400% | −3.0550% |
| | 2031–2035 | 0.3300% | 3.5100% | −5.2300% | −8.1400% | −3.0600% |

4.5.2. Basis for Setting Indicators in Tianjin

The indicators of Tianjin are set in Table 6. According to the Tianjin Bureau of Statistics data, by the end of 2021, Tianjin's permanent population had reached 15.6966 million, an increase of 0.4% over the previous year. Since 2016, the resident population of Tianjin has been showing a steady growth trend. Among them, Tianjin's population growth rate dropped slightly in 2020 due to the epidemic. However, with the effective control of the epidemic, Tianjin's population growth rate rose again in 2021. The population will continue to grow in the future as Tianjin's economy continues to develop. According to the plan of the Tianjin municipal government, the permanent population of Tianjin will reach about 20 million by 2035. In 2019, the resident population of Tianjin was 13.85 million. The annual change rate from 2019–2035 will be controlled at about 2.7753%. Then, it can be set that in 2021, the low speed of Tianjin's resident population can be set as 2.5800%, the medium speed can be set as 2.7753%, and the high speed can be set as 2.9800%, with an even annual decline of 0.003% during the forecast period. Gross regional product per capita can be set to be RMB 54,053 per capita in Tianjin in 2010 and RMB 101,557 per capita in Tianjin in 2019; it can be calculated that the average annual rate of change from 2019-2035 will be controlled at about 8.79%. The GDP per capita in 2021 can be set at 8.5884% at low speed, 8.7884% at medium speed, and 8.9884% at high speed. Energy consumption intensity in Tianjin was 1.00% in 2010 and 0.58% in 2019, so the change rate between 2010 and 2019 can be calculated as −4.15%. Energy consumption intensity in 2021 can be set to −4.4462% at low speed, −4.1462% at medium speed, and −3.8462% at high speed. The energy structure in Tianjin was 0.71% in 2010 and 0.46% in 2019, indicating a change rate of −3.52% between 2010 and 2019. The energy structure can be set at −3.8178% for low speed, −3.5178% for medium speed, and −3.2178% for high speed in 2021. In Tianjin, the proportion of the secondary industry in GDP was 52.5% in 2010 and 35.2% in 2019, so the change rate between 2010 and 2019 can be calculated as −3.30%. The proportion of secondary industry in GDP in 2021 can be set as −3.492% at low speed, −3.2957% at medium speed, and −3.0952% at high speed.

4.5.3. Basis for Setting the Indicators in Hebei

The setting of the indicators for Hebei is shown in Table 7. Population setting: According to the Population Development Plan of Hebei Province (2018–2035), the population of Hebei Province will increase to 79.1 million people by 2035. In 2019, the resident population of Hebei Province was 21.9 million people; it can be calculated that the average annual rate of change from 2019–2035 will be controlled at about 0.39%. The population can be set to

have a low speed of 0.3636%, a medium speed of 0.3886%, and a high speed of 0.4086% in 2021, with a predicted annual average decrease of 0.01% during the forecast period. The per capita GDP of Hebei Province was 25,308 yuan in 2010 and 47,036 yuan in 2019, and the change rate from 2010–2019 can be calculated as about 8.59%. It can be set that the per capita regional gross domestic product in 2021 will decrease at a low speed of 8.2854%, a medium speed of 8.5854%, and a high speed of 8.7857%. During the forecast period, the annual average decline will be 0.02%. Energy consumption intensity in Hebei was 1.53% in 2010 and 0.93% in 2019; therefore, the rate of change between 2010–2019 can be calculated to be −3.94%, which can be set as −3.9674% for the low rate of energy consumption intensity in 2021, −3.9374% for the medium rate, and -3.9074% for the high rate. Energy structure in Hebei was 1.00% in 2010 and 0.88% in 2019; then, the rate of change between 2010–2019 can be calculated as −1.15% and the energy structure in 2021 can be set to be −1.5483% for the low rate, −1.1438% for the medium rate, and −0.7483% for the high rate. In 2010, the share of secondary industry in the GDP in Hebei was 52.5%, and in 2019, the share of secondary industry in the GDP in Hebei Province was 38.7%; then, the rate of change between 2010–2019 can be calculated as −2.63%, and it can be set that, in 2021, the share of secondary industry in GDP will be −2.9286% at low speed, −2.6286% at medium speed, and −2.3286% at high speed.

**Table 6.** Tianjin indicator settings.

| Rate of Change | Vintages | Rate of Change Setting | | | | |
| | | Permanent Population | GDP per Capita | Intensity of Energy Consumption | Energy Mix | Share of Secondary Sector in GDP |
| --- | --- | --- | --- | --- | --- | --- |
| Low | 2019–2020 | −1.0000% | 6.8772% | 7.3418% | −3.1827% | −4.2122% |
| | 2021–2025 | 2.5800% | 8.5884% | −4.4462% | −3.8178% | −3.4952% |
| | 2026–2030 | 2.4300% | 5.0884% | −4.4962% | −1.3178% | −3.4957% |
| | 2031–2035 | 2.2800% | 1.5884% | −4.5462% | 1.1822% | −3.4962% |
| Middle | 2019–2020 | −1.0000% | 6.8772% | 7.3418% | −3.1827% | −4.2122% |
| | 2021–2025 | 2.7753% | 8.7884% | −4.1462% | −3.5178% | −3.2952% |
| | 2026–2030 | 2.6253% | 5.2884% | −4.1962% | −1.0178% | −3.2957% |
| | 2031–2035 | 2.4753% | 1.7884% | −4.2462% | 1.4822% | −3.2962% |
| High | 2019–2020 | −1.0000% | 6.8772% | 7.3418% | −3.1827% | −4.2122% |
| | 2021–2025 | 2.9800% | 8.9884% | −3.8462% | −3.2178% | −3.0952% |
| | 2026–2030 | 2.8300% | 5.4884% | −3.8962% | −0.7178% | −3.0957% |
| | 2031–2035 | 2.6800% | 1.9884% | −3.9462% | 1.7822% | −3.0962% |

### 4.5.4. Basis for Setting the Indicators in Henan

The setting of each indicator in Henan is shown in Table 8. In the population setting, the resident population of Henan Province in 2010 was 94.05 million people, and the resident population of Henan Province in 2019 was 99.01 million. Then, the rate of change between 2010 and 2019 can be calculated as 0.53%, and it can be set that the resident population in 2021 will be 0.4974% at a low rate, 0.5274% at a medium rate, and 0.5574% at a high rate. The uniform rate of decline is 0.001 percent per year over the projection period. The per capita GDP of Henan Province was 41,326 yuan in 2016 and 5,450 yuan in 2019. The annual change rate from 2019–2035 will be controlled at about 7.88%. It can be set that the per capita regional gross domestic product in 2021 will decrease at a low speed of 7.5824%, a medium speed of 7.8824%, and a high speed of 8.1824%, with a predicted average annual decrease of 0.001% during the forecast period. Energy consumption intensity in Henan Province was 0.95% in 2010 and 0.41% in 2019. Then, the rate of change between 2010 and 2019 can be calculated as −5.66%, and it can be set that the low rate of energy consumption intensity in 2021 will be −5.9567%, the medium rate will be −5.6567% and the high rate will be −5.3567%. The energy structure of Henan Province was 1.22% in 2010 and 0.90% in 2019; then, the rate of change between 2010 and 2019 can be calculated as −2.60%, and

it can be set that the low rate of energy structure in 2021 can be set to be −2.9025%, the medium rate can be set to be −2.6025%, and the high rate can be set to be −2.3025%. In Henan Province, the share of secondary industry in GDP was 57.30% in 2010 and 43.50% in 2019. Then, the rate of change between 2010 and 2019 can be calculated to be −2.41%, and it can be set that the share of secondary industry in GDP in 2021 will be −2.7084% at low speed, −2.4084% at medium speed, and −2.1084% at high speed.

**Table 7.** Setting of indicators in Hebei.

| Rate of Change | Vintages | Rate of Change Setting | | | | |
| | | Permanent Population | GDP per Capita | Intensity of Energy Consumption | Energy Mix | Share of Secondary Sector in GDP |
|---|---|---|---|---|---|---|
| Low | 2019–2020 | 0.2441% | 5.3944% | −1.5503% | −2.4988% | −4.6629% |
| | 2021–2025 | 0.3686% | 8.2854% | −3.9674% | −1.5483% | −2.9286% |
| | 2026–2030 | 0.3186% | 8.3854% | −4.1674% | −2.0483% | −3.1286% |
| | 2031–2035 | 0.2686% | 8.4854% | −4.3674% | −2.5483% | −3.3286% |
| Middle | 2019–2020 | 0.2441% | 5.3944% | −1.5503% | −2.4988% | −4.6629% |
| | 2021–2025 | 0.3886% | 8.5854% | −3.9374% | −1.1483% | −2.6286% |
| | 2026–2030 | 0.3836% | 8.6854% | −4.1374% | −1.6483% | −2.8286% |
| | 2031–2035 | 0.3786% | 8.7854% | −4.3374% | −2.1483% | −3.0286% |
| High | 2019–2020 | 0.2441% | 5.3944% | −1.5503% | −2.4988% | −4.6629% |
| | 2021–2025 | 0.4086% | 8.7854% | −3.9074% | −0.7483% | −2.3286% |
| | 2026–2030 | 0.4036% | 8.8854% | −4.1074% | −1.2483% | −2.5286% |
| | 2031–2035 | 0.3986% | 8.9854% | −4.3074% | −1.7483% | −2.7286% |

**Table 8.** Setting of Indicators in Henan.

| Rate of Change | Vintages | Rate of Change Setting | | | | |
| | | Permanent Population | GDP per Capita | Intensity of Energy Consumption | Energy Mix | Share of Secondary Sector in GDP |
|---|---|---|---|---|---|---|
| Low | 2019–2020 | 0.3145% | 7.8824% | −6.5153% | −3.7995% | −2.1667% |
| | 2021–2025 | 0.4974% | 7.5824% | −5.9567% | −2.9025% | −2.7084% |
| | 2026–2030 | 0.4924% | 7.5874% | −6.1567% | −3.4025% | −2.9084% |
| | 2031–2035 | 0.4874% | 7.5924% | −6.3567% | −3.9025% | −3.1084% |
| Middle | 2019–2020 | 0.3145% | 7.8824% | −6.5153% | −3.7995% | −2.1667% |
| | 2021–2025 | 0.5274% | 7.8824% | −5.6567% | −2.6025% | −2.4084% |
| | 2026–2030 | 0.5224% | 7.8874% | −5.8567% | −3.1025% | −2.6084% |
| | 2031–2035 | 0.5174% | 7.8924% | −6.0567% | −3.6025% | −2.8084% |
| High | 2019–2020 | 0.3145% | 7.8824% | −6.5153% | −3.7995% | −2.1667% |
| | 2021–2025 | 0.5574% | 8.1824% | −5.3567% | −2.3025% | −2.1084% |
| | 2026–2030 | 0.5524% | 8.1874% | −5.5567% | −2.8025% | −2.3084% |
| | 2031–2035 | 0.5474% | 8.1924% | −5.7567% | −3.3025% | −2.5084% |

*4.6. Scenario Building*

Based on the impact of the high, medium, and low three rates of change in the four provinces and cities, five scenarios were constructed to carry out the five scenarios for 2020–2035 in each of the four provinces and cities. The results are shown in Table 9.

Low-carbon development scenario (M1): The rates of change between the indicators in this scenario are chosen to be low, exploring the impact of indicators on carbon emissions in a lower scenario. Energy efficiency scenario (M2): Only the resident population and GDP per capita are changed in the low-carbon development scenario, while the rest of the indicators remain unchanged. In the energy-saving scenario, the industrial structure has been optimized, energy consumption has decreased, and the proportion of the secondary

industry to GDP has decreased. Baseline scenario (M3): The median value of changes among the indicators is adopted, and the influence of the original policy intensity on carbon emissions is analyzed without adjustment. Ideal scenario (M4): The ideal scenario has an increase in the resident population, a significant increase in GDP per capita, minimum energy consumption between categories, and green and energy-efficient development. Free development scenario (M5): In this scenario, the variation between the indicators is the highest, and there is no influence from other factors, only from development.

**Table 9.** Peak carbon forecast scenarios for four provinces and cities, 2020–2035.

| Scenarios | Permanent Population | GDP per Capita | Intensity of Energy Consumption | Energy Mix | Share of Secondary Sector in GDP |
|---|---|---|---|---|---|
| Low-carbon development scenario (M1) | low | low | low | low | low |
| Energy efficiency scenario (M2) | middle | middle | low | low | low |
| Baseline scenario (M3) | middle | middle | middle | middle | middle |
| Ideal scenario (M4) | middle | high | low | low | low |
| Freedom to develop scenario (M5) | high | high | high | high | high |

### 4.7. Predictive Analyses

From (Figure 6): Beijing and Henan peaked in the 2000–2035 timeframe under the analysis of different scenarios, with Beijing peaking in 2010 and Henan peaking in 2011. Carbon emissions continued to decline in two regions, Beijing and Henan, over the next 2035 period. The Tianjin data indicated an upward trend in the 2000–2019 timeframe, with a downward trend in carbon emissions under the low-carbon development scenario model over the projection period. Hebei Province did not experience carbon peaking between 2020 and 2035, with an inflection point in 2030 under the low-carbon development model and an upward trend in carbon emissions under the remaining four models.

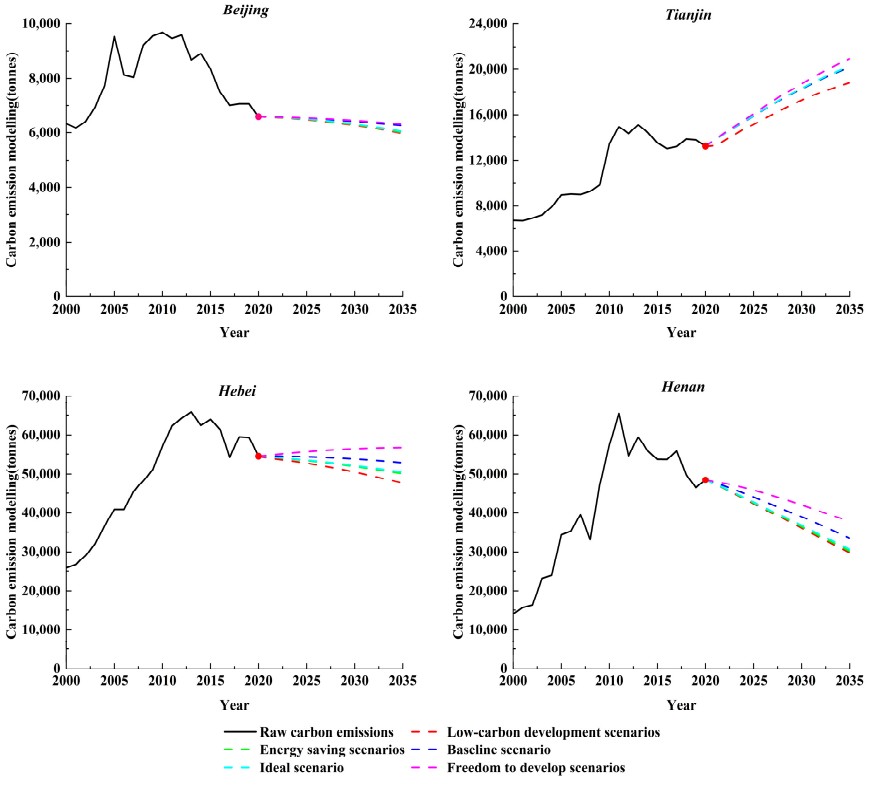

**Figure 6.** Carbon emission projections for four provinces and municipalities under various scenarios, 2020–2035 ($10^4$t).

## 5. Discussion

Wang Wenju et al. analyzed national and provincial governments' overall objectives and critical action plans to promote carbon peaking. They used the Mann–Kendall statistical trend test to examine each province's carbon emission peaking situation [30]. The study found that provincial governments have responded positively to the central government's request to formulate and actively implement action programs for carbon peaking. Beijing and Henan have taken the lead in achieving carbon peaking. Provinces have implemented exceptional government support and financial subsidies, effectively guiding low-carbon development, reducing carbon emissions, and thus achieving carbon peaks. Han Nan et al. constructed a carbon emission system dynamics model by analyzing the relationship between carbon emissions and influencing factors. They set up six scenarios to simulate and predict their impact on the time of carbon peaking in Beijing, Tianjin, and Hebei [31]. The results showed that under the baseline scenario, Beijing has already achieved peak carbon. Tianjin is expected to reach peak carbon by 2023, and Hebei is having difficulty reaching peak carbon by 2035. This is consistent with the findings of this paper.

Our study suggests that Hebei is less likely to reach peak carbon emissions in 2035. However, Beijing, Tianjin, and Henan could reach peak carbon emissions in the 2020–2030 timeframe. Beijing is actively promoting the construction of a green Beijing, deepening the implementation of the functional positioning of the capital city, and taking the lead in establishing the development concept of reduction. Tianjin actively promotes the development of a digital economy, the transformation and upgrading of traditional industries, and the gradual decline of carbon emissions. Henan vigorously promotes energy conservation and emission reduction, accelerates the establishment of a sound economic system of green, low-carbon, and recycling development, promotes the overall green and low-carbon economic and social transformation, and helps to achieve the goal of carbon peak and carbon neutrality. Hebei has formed a diversified pillar industry pattern featuring resource-consuming and polluting industries such as iron and steel, coal, chemicals, and equipment manufacturing. The increased $CO_2$ emissions from the excessive use of fossil energy sources, such as coal, have put enormous pressure on environmental protection. From the study of decoupling effects, the relationship between economic growth and carbon emissions in provinces and cities along the South-to-North Water Diversion Central Route Project has been generally improving in recent years, and a more desirable decoupling will be achieved in the future.

As of 22 July 2022, the water entering the central canal from the Taocha Canal Headwork of the first phase of the South-to-North Water Diversion Project exceeded 50 billion cubic meters, benefiting a population of more than 85 million. The annual volume of water transferred by the first phase of the Central Route Project has continued to climb from more than 2 billion cubic meters to 9 billion cubic meters. It demonstrates that the South-to-North Water Diversion Project continues to develop highly, providing high-quality water security for the provinces and cities along the route.

In order to reduce energy consumption and achieve carbon reduction goals in the South-to-North Water Diversion Project, consolidating the foundation of green and low-carbon management is fundamental. We should strengthen compliance management, continuously improve green development systems such as environmental protection, pollution control, energy and resource conservation, and efficient utilization, and low-carbon transformation, and guide the implementation of standards and requirements that are conducive to green development. We should also establish and improve a long-term mechanism for green development, actively explore the establishment of effective incentive and constraint mechanisms, promote innovation in green development management and institutional innovation, dynamically monitor energy consumption and carbon emissions, achieve monitoring, reporting, and verification of energy consumption and carbon emissions indicators, and provide decision-making support for green and low-carbon development. Provinces and municipalities along the route should formulate energy-saving and emission-reduction

policies suitable for their provinces according to local conditions to contribute to China's goal of achieving carbon peaking by 2030.

### 6. Conclusions

This study uses the STIRPAT model to investigate the carbon emissions of provinces and cities along the South-to-North Water Diversion Central Route Project. It also forecasts carbon emissions from 2020–2035 under different scenarios and analyses whether carbon can be peaked by 2035. The main conclusions are as follows:

(1) The four provinces and municipalities were mainly weakly decoupled in the 2000–2009 timeframe, gradually shifting to strong decoupling from 2010–2019. From the perspective of decoupling economic development from carbon emissions, a country or region usually goes through a process progressing from negative decoupling to weak or even strong decoupling. Moreover, the process is often tortuous. For example, recessionary decoupling and negative recessionary decoupling can occur under the influence of political, economic, and environmental factors.

(2) According to the parameters of the model formula for the four provinces and cities, it can be seen that the resident population and per capita GDP have a more significant impact on carbon emissions. Due to this, Beijing's resident population and per capita GDP can cause a change of 0.2561% and 0.1878% for every 1% change. Every 1% change in Tianjin's resident population and GDP per capita can cause a change of 1.0539% and 0.1522%.

(3) There will be carbon peaks in both Beijing and Henan in the 2000–2035 timeframe, with Beijing peaking at 96.836 million tons in 2010 and Henan peaking at 654.104 million tons in 2011. Mainly, the continued optimization of the industrial structure, promoting a clean energy transition, and implementing the Peak Carbon Implementation Program will achieve the Peak Carbon Goal on schedule.

(4) Among the four provinces and cities along the South-to-North Water Diversion Project, only Hebei did not reach its peak during the period under study, which is related to a large amount of energy consumption in Hebei, a traditional industrial province. The total energy consumption in Hebei Province is significant, and its structure is dominated by fossil energy. The proportions of coal consumption per capita and GDP energy consumption are higher than the national level, resulting in more significant carbon emissions.

**Author Contributions:** Q.M.: Supervision, Project administration, Funding acquisition, Resources. B.L.: Conceptualization, Data curation, Methodology, Software, Visualization, Formal analysis, Writing review & editing. Y.Z.: Writing original draft, Writing review & editing. H.Z.: Investigation, Resources, Writing original draft. Z.X.: Methodology, Validation, Investigation. Y.L.: Writing original draft, Methodology. Q.L.: Data curation, Visualization. All authors have read and agreed to the published version of the manuscript.

**Funding:** This work was supported by the National Natural Science Foundation of China (Project No.: 42371314), the Research Project of Henan Federation of Social Sciences in 2021 (Project No.: SKL-2021-3196), the Research Project of Humanities and Social Sciences in Colleges and Universities of Henan Province (Project No.: 2021-ZZJH-157), and the Philosophy and Social Science Program of Henan Province (Project No.: 2019CJJ078).

**Data Availability Statement:** The datasets used and/or analyzed during the current study are available from the corresponding author on reasonable request.

**Conflicts of Interest:** The authors declare no conflict of interest.

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
