# Peer review of "Multi-Scenario Prediction Analysis of Carbon Peak Based on STIRPAT Model-Take South-to-North Water Diversion Central Route Provinces and Cities as an Example"

_land, doi:10.3390/land12112035_

Round 1
Reviewer 1 Report
Comments and Suggestions for Authors
This manuscript addresses an important topic: carbon emissions and the related subject of "decoupling" them with economic growth. The problem is that the manuscript is very poorly written, so much so that it was difficult for me to make it through to the end.
"Gradually becoming one of the major constraints to economic development." Near the beginning of the abstract, this collection of words is what the authors pass off for a sentence. This is not an isolated case. I am not going through the entire manuscript grammatically for this review, but I will point out a couple of other examples before proceeding to the substance.
In the small section that begins on line 172, it looks like the authors are confused between using a period to end a sentence or to use a comma to continue the sentence. Why else would "Energy mix" and "The share" be capitalized when they come after commas? The way these and many other sentences are written are unacceptable.
The authors go out of their way on lines 238-241 to make sure the reader knows what a Pearson correlation coefficient is, which is something all the readers of this journal already know, but at the same time glosses over Bartlett's sphericity test and KMO sampling procedures, items I have never heard of. This shows a very poor balance in the presentation of method.
In Table 3, the authors have a word that says: Spproximate
I like the idea of using regression to identify and measure decoupling, etc. I notice the authors went from straight multiplication of terms in the IPAT model to using exponents and then a log transformation. I saw STIRPAT as early as the abstract, but I do not recall seeing how the acronym was derived.
On lines 273 to 277 the authors state that in order to avoid multicollinearity, they used a ridge regression. I do not recall seeing a test for multicollinearity in the manuscript. Why should there be collinearity between the independent variables, if they are truly independent? Seeking a technical fix such as ridge regression should be the last thing a researcher does, because if the independent variables are correlated, that should be telling us something. Are they related in a time series sent, where we have strong trends in all the variables? How does ridge regression solve this problem?
From this point on in the results, the amount of statistics presented is enormous. It is very difficult to get through. Is it necessary to present all these numbers to derive the conclusions the authors get in sections 5 and 6?
Those two sections seem to be consistent with the empirical findings.
In order to be published in Land, this manuscript needs a lot of work, from a complete English language editing and revision, to a more thorough description of terms used in Methods. But I like the concept of decoupling and attempting to measure it empirically.
Decision: Major Revision
Comments on the Quality of English Language
The English needs a lot of work. Many of the "sentences" are not complete sentences at all. They must be rewritten. This manuscript needs a thorough English language edit and revision. I have some more details in the comments to authors and to the editor.
Author Response
Introduction
We feel great thanks for your professional review work on our article. As you are concerned, there are several problems that need to be addressed. According to your nice suggestions, we have made extensive corrections to our previous draft, the detailed corrections are listed below.
1.*This manuscript addresses an important topic: carbon emissions and the related subject of "decoupling" them with economic growth. The problem is that the manuscript is very poorly written, so much so that it was difficult for me to make it through to the end.
Response: The text has been revised with regard to the decoupling of four provinces and municipalities, namely Beijing, Tianjin, Hebei and Henan, for the period 2000-2019. Figure 3 has been modified.
2.*"Gradually becoming one of the major constraints to economic development." Near the beginning of the abstract, this collection of words is what the authors pass off for a sentence. This is not an isolated case. I am not going through the entire manuscript grammatically for this review, but I will point out a couple of other examples before proceeding to the substance.
Response: The example "On 22 September 2020, President Xi Jinping announced at the seventy-fifth session of the United Nations General Assembly that China will strive to peak its carbon dioxide emissions by 2030 and work towards achieving carbon neutrality by 2060" has been added to illustrate the importance of carbon dioxide emissions. In this context, the carbon emissions of the provinces and cities along the South-to-North Water Diversion Route are discussed.
3.* In the small section that begins on line 172, it looks like the authors are confused between using a period to end a sentence or to use a comma to continue the sentence. Why else would "Energy mix" and "The share" be capitalized when they come after commas? The way these and many other sentences are written are unacceptable.
Response: The use of punctuation in the text has been revised, please check it out.
4.*The authors go out of their way on lines 238-241 to make sure the reader knows what a Pearson correlation coefficient is, which is something all the readers of this journal already know, but at the same time glosses over Bartlett's sphericity test and KMO sampling procedures, items I have never heard of. This shows a very poor balance in the presentation of method.
Response: The KMO test metrics have been modified accordingly in the text, as well as the textual narrative has been supplemented.
5.*In Table 3, the authors have a word that says: Spproximate
Response: Writing errors have been corrected in the text, please check it out.
6.*I like the idea of using regression to identify and measure decoupling, etc. I notice the authors went from straight multiplication of terms in the IPAT model to using exponents and then a log transformation. I saw STIRPAT as early as the abstract, but I do not recall seeing how the acronym was derived.
Response: Changes have been made in the text. This paper focuses on using the extended STIRPAT model, The logarithmic treatment of IPAT is mainly to facilitate subsequent data analysis and processing. IPAT is the base form of the STIRPAT model
7.* On lines 273 to 277 the authors state that in order to avoid multicollinearity, they used a ridge regression. I do not recall seeing a test for multicollinearity in the manuscript. Why should there be collinearity between the independent variables, if they are truly independent? Seeking a technical fix such as ridge regression should be the last thing a researcher does, because if the independent variables are correlated, that should be telling us something. Are they related in a time series sent, where we have strong trends in all the variables? How does ridge regression solve this problem?
Response: Has been changed in the text. In this paper, the use of SPSS software test concluded that the variance inflation factor between the variables are greater than 10 there is multicollinearity, need to use ridge regression to solve the problem of covariance. When there is a strong multicollinearity between the independent variables, the resulting multiple linear regression model is very unstable; the ridge regression analysis can solve this problem well.
8.* From this point on in the results, the amount of statistics presented is enormous. It is very difficult to get through. Is it necessary to present all these numbers to derive the conclusions the authors get in sections 5 and 6?
Response: Has been changed in the text. Which form the basis for the 2020-2035 multi-scenario modelling. The indicators are mainly derived from the 14th Five-Year Plan and the 2020-2035 Vision of the municipalities. This is the basis for adjusting the parameters and indicators on which these data are very important to be used for carbon emission projections.
9.*Those two sections seem to be consistent with the empirical findings.
Response: The conclusions and some inappropriate parts of the discussion in the paper have been revised, and the projections of future carbon emissions can provide a policy basis for the government.

Reviewer 2 Report
Comments and Suggestions for Authors
The author has done a lot of data collection and analysis work.Carbon emission data from China Carbon Emission Database 2000-2019. The panel data of statistical yearbooks of 4 provinces and cities were analyzed by Tapio decoupling model. SPSS was used to perform KMO and Bartlett's test with correlation for indicator selection. The extended STIRPAT model and ridge regression were used for analysis. and projections of carbon peaks in the study area for the years 2020-2035 under different rates of change simulated by the scenario analysis method.These works are very good and systematic, but I believe there are still the following issues that need to be addressed
1.The title is:Multi-Scenario Prediction Analysis of Carbon Peak Based on STIRPAT Model-Take South-to-North Water Diversion Central Route Provinces and Cities as an Example,The article lacks comparative discussion before and after the implementation of the project, and the content is different from the topic
2.As the middle line of the South to North Water Diversion Project, the future carbon emissions of the South to North Water Diversion Project are directly related to the water diversion volume. However, the article lacks data and discussion on this part, and only analyzes based on the data of each province, lacking macro thinking and overall analysis
3.In order to reduce energy consumption and achieve carbon reduction goals in the South to North Water Diversion Project, consolidating the foundation of green and low-carbon management is fundamental. We should strengthen compliance management, continuously improve green development systems such as environmental protection, pollution control, energy and resource conservation and efficient utilization, and low-carbon transformation, and guide the implementation of standards and requirements that are conducive to green development; Establish and improve a long-term mechanism for green development, actively explore the establishment of effective incentive and constraint mechanisms, promote innovation in green development management and institutional innovation, dynamically monitor energy consumption and carbon emissions, achieve monitoring, reporting, and verification of energy consumption and carbon emissions indicators, and provide decision-making support for green and low-carbon development. Suggest the author to discuss the conclusions based on these issues
Comments on the Quality of English LanguageModerate editing of English language required
Author Response
Introduction
We feel great thanks for your professional review work on our article. As you are concerned, there are several problems that need to be addressed. According to your nice suggestions, we have made extensive corrections to our previous draft, the detailed corrections are listed below.
1.The title is: Multi-Scenario Prediction Analysis of Carbon Peak Based on STIRPAT Model-Take South-to-North Water Diversion Central Route Provinces and Cities as an Example, The article lacks comparative discussion before and after the implementation of the project, and the content is different from the topic.
Response: Changes have been made in the corresponding parts of the text. On 30 December 2003, the South-North Water Diversion Project was officially launched. On 12 December 2014 the first phase of the South-to-North Water Diversion Project was officially opened. The study period of this paper is 2000-2019 and the forecast period is 2020-2035, and the study is before and after the implementation of the whole project is included.
2.As the middle line of the South to North Water Diversion Project, the future carbon emissions of the South to North Water Diversion Project are directly related to the water diversion volume. However, the article lacks data and discussion on this part, and only analyzes based on the data of each province, lacking macro thinking and overall analysis
Response: In the discussion section of the article, changes in the volume of water transferred from the main canal, and the annual volume of water transferred from the first phase of the Central Route Project have been included. Used to describe a reduction in carbon emissions and an increase in water transfers for high-quality development outcomes, indicating continued positive development.
3.In order to reduce energy consumption and achieve carbon reduction goals in the South to North Water Diversion Project, consolidating the foundation of green and low-carbon management is fundamental. We should strengthen compliance management, continuously improve green development systems such as environmental protection, pollution control, energy and resource conservation and efficient utilization, and low-carbon transformation, and guide the implementation of standards and requirements that are conducive to green development; Establish and improve a long-term mechanism for green development, actively explore the establishment of effective incentive and constraint mechanisms, promote innovation in green development management and institutional innovation, dynamically monitor energy consumption and carbon emissions, achieve monitoring, reporting, and verification of energy consumption and carbon emissions indicators, and provide decision-making support for green and low-carbon development. Suggest the author to discuss the conclusions based on these issues.
Response: This paper updates national and international research advances and provides a more comprehensive analysis of the limitations of previous studies.
Round 2
Reviewer 1 Report
Comments and Suggestions for Authors
The authors made a lot of very good improvements to the manuscript. Lots of good corrections - minor revision required by way of thorough English editing still needed.
Comments on the Quality of English Language
The authors made some improvements, but I am asking the editors to just take a moment to look at the grammar. Early in the abstract, the authors have this as a sentence:
In order to ensure that the "peak carbon" and "carbon neutral" targets are met on time.
This is not a complete sentence. It is a sentence fragment, also known as a clause. Please go over the manuscript again and ensure that all the grammar is correct. The authors made an improvement but the English still needs work. In that example, a complete sentence would be a comma after 2060 and take the capital off of In order to ensure........
Right after that sentence the authors say "coordinate the coordinated development..." This is not acceptable English. Just delete coordinated and write "coordinate the development" or "how to assure the coordinated development" You just cannot use two forms of the same word together like the authors have done. "Coordinate the coordinated development" is not acceptable English. Please correct that.
An editor needs to go through this entire manuscript with the authors to check for these mistakes. Another problem is that carbon should not be capitalized in the first sentence. Why would the authors do that?
Please remember what an important piece of work you are publishing here. It is something that will endure in the literature well into the future. You do not want to be remembered for minor grammar mistakes. You want to be remembered for writing an important piece that stands the test of time and shows you really made a scholarly contribution.
I gave some examples in the Abstract.
Author Response
Introduction
We feel great thanks for your professional review work on our article. As you are concerned, there are several problems that need to be addressed. According to your nice suggestions, we have made extensive corrections to our previous draft, the detailed corrections are listed below.
1.*In order to ensure that the "peak carbon" and "carbon neutral" targets are met on time. This is not a complete sentence. It is a sentence fragment, also known as a clause. Please go over the manuscript again and ensure that all the grammar is correct. The authors made an improvement but the English still needs work. In that example, a complete sentence would be a comma after 2060 and take the capital off of In order to ensure........
Response: About the appearance of what is not a complete sentence, problems with initial capitalization, and incorrect use of punctuation, which have been corrected in the text. Please check it out.
2.*Right after that sentence the authors say "coordinate the coordinated development..." This is not acceptable English. Just delete coordinated and write "coordinate the development" or "how to assure the coordinated development" You just cannot use two forms of the same word together like the authors have done. "Coordinate the coordinated development" is not acceptable English. Please correct that.
Response: Many thanks for the valuable suggestions; changes have been made in the text; please check it out.
